# A New Method for Friction Estimation in EMA Transmissions

Gaetano Quattrocchi , Alessandro Iacono, Pier C. Berri , Matteo D. L. Dalla Vedova * and Paolo Maggiore

Department of Aerospace and Mechanical Engineering (DIMEAS), Politecnico di Torino, 10129 Turin, Italy; gaetano.quattrocchi@polito.it (G.Q.); alessandro.iacono@studenti.polito.it (A.I.); pier.berri@polito.it (P.C.B.); paolo.maggiore@polito.it (P.M.)
* Correspondence: matteo.dallavedova@polito.it

**Abstract:** The increasing interest for adopting electromechanical actuators (EMAs) on aircraft demands improved diagnostic and prognostic methodologies to be applied to such systems in order to guarantee acceptable levels of reliability and safety. While diagnostics methods and techniques can help prevent fault propagation and performance degradation, prognostic methods can be applied in tandem to reduce maintenance costs and increase overall safety by enabling predictive and condition-based maintenance schedules. In this work, a predictive approach for EMAs friction torque estimation is proposed. The algorithm is based on the reconstruction of the residual torque in mechanical transmissions. The quantity is then sampled and an artificial neural network (ANN) is used to obtain an estimation of the current health status of the transmission. Early results demonstrate that such an approach can predict the transmission health status with good accuracy.

**Keywords:** prognostics; electromechanical actuators; neural network; friction





## 1. Introduction

Electromechanical Actuators (EMAs) offer great advantages over their traditional counterparts (namely old hydromechanical and modern electrohydraulic actuators) when used as actuation devices on aircraft. They represent the natural evolution of actuation systems in the *more electric* and *all electric* aircraft design philosophies [1] since using EMAs for both primary and secondary flight controls would eliminate the need for hydraulic and pneumatic power aboard the aircraft, leading to overall weight reduction and a more convenient method to distribute mechanical power across the aircraft: distributing electrical power directly to the end users is easier and lighter than distributing pressurized hydraulic fluid.

Still, as of today, the use of EMAs is limited to secondary flight control (such as airbrakes, spoilers and high-lift devices) on large aircraft, and they are used as primary flight control actuators only on small UAVs and, in general, applications where the loss of the actuation system is neither mission critical nor would result in a loss of life or loss of expensive flying systems. This is partially explained by the fact that EMAs are still a relatively new technology in the aerospace sector: their combined fault modes have yet to be fully understood, and they generally lack established prognostic methodologies.

Nonetheless, in recent years, many diagnostic and prognostic methods for EMAs have been proposed. The aim of prognostic methods is the estimation of the health status and/or the Remaining Useful Life (RUL) of various components of the EMA [2] so that they can be isolated or replaced accordingly; this is a cardinal principle of modern Prognostics and Health Management (PHM) philosophies [3].

PHM starts with Fault Detection and Identification (FDI) [4]. In this phase, early signs of developing faults are identified by analyzing the response of the system. RUL is then estimated by using information about the status of the system acquired at this stage.

FDI strategies are classified into two main categories, namely *model-based* and *data-driven* techniques, which, aside from being best suited for real time or offline FDI, differ on a fundamental level in their approaches.

Model-based FDI is based on the idea of comparing the actual response of the system with that of a simulation model used for health monitoring either in time [2,5–9] or frequency domain [10–13], leveraging a fast Fourier Transformation [14,15] in the latter. Other applications include the use of extended Kalman filters for evaluating friction and, thus, faults in EMA such as in [16]. In any case, an accurate modeling of EMA is very important; multi-domain interactions and faults need to be included in the model as in [17].

On the other hand, data-driven methods for FDI substitute detailed physical knowledge of the system for large databases of logged system outputs. Such data are then used in machine learning algorithms for estimating the health status of the real system. To this end, data-driven methods include the use of Artificial Neural Networks (ANNs) of various types [18–20], such as LSTMs [21], convolutional neural networks [22], fuzzy logic [23] or other machine learning algorithms [24].

Other proposed methods for estimating the health status of systems rely on the use of machine learning algorithms for analyzing one or more signals outputted by the system or reconstructed from output variables (as in the case of the back-electromotive force or BEMF), which are considered prognostic indicators; this approach is often described as hybrid since it leverages both machine learning techniques and knowledge of the physical system. The analysis is, thus, performed with specifically trained ANNs that may be trained on data obtained from simulated models, which use said prognostic indicators in order to estimate the health status of one or more components [25,26].

In this framework, the residual torque, defined as the sum of all the friction and viscous torques in the transmission of the actuator, stands out as a possible candidate for being a valid indicator as it carries information about the friction torques (variations are a telling indicator of wearing, possible jamming and other kinds of degradation in the transmission) of the system and can be reconstructed from other data acquired during the functioning of the EMA, such as the electrical current in the motor, the acceleration of the shaft and the hinge moment on the actuator. In this work, the viability of the residual torque as a prognostic indicator for EMAs in a neural-network-based methodology is investigated, both in the context of a pre/post-flight routine on ground and of real time use during the flight. The static and dynamic friction torques, as well as their ratio and the transmission efficiency under both aiding and opposing loads, are the considered targets of interest for this application.

Figure 1 shows a schematization of the proposed prognostic method. This work focuses on the upper branch of the flowchart.

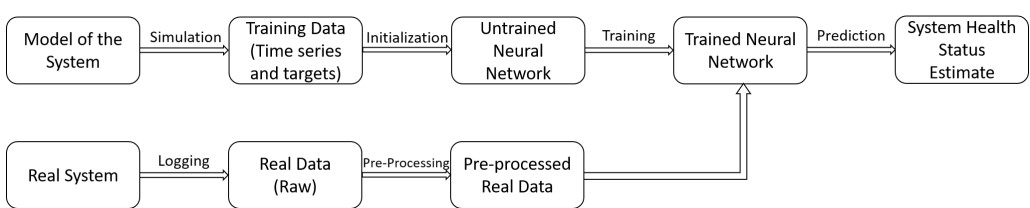

**Figure 1.** Schematization of the proposed method.

*Scope of the Work*

The scope of this work is concerned with applying the prognostic methodology for EMAs proposed in [25] by using the reconstructed signal for residual torque during actuation as the main prognostic indicator or "mapping signal". The methodology consists in equipping the real-life system with sensors that can measure some quantities which are then used to reconstruct other quantities that are used as inputs of an ANN.

For instance, the rotational speed of an electric motor can easily be measured by using a Hall effect sensor, while the equivalent dynamic friction torques of the transmission linked to said electric motor, for which its deviation from the nominal value holds precious information about the health status of the transmission itself, generally requires partial disassembly and characterization of the system.

In this paper, the idea is to link easy-to-measure quantities (such as currents, voltages, rotor angular position and speed) to the health status of the system. In order to conduct this, the residual torque, a synthetic quantity, is calculated using a virtual sensor approach, and this quantity is then fed to an ANN to obtain a punctual estimation of the dry friction torques of the mechanical transmission.

## 2. Materials and Methods

### 2.1. System Overview

The actuator model of the system used for the simulations in this work is derived from [27] and will be briefly described. The top level of the model is shown in Figure 2. The model is composed of a COM block that acts as a signal generator, a model of the EMA itself and the model of the longitudinal dynamic of an aircraft. All the other blocks are used to monitor and log the response of the system into a MATLAB file.

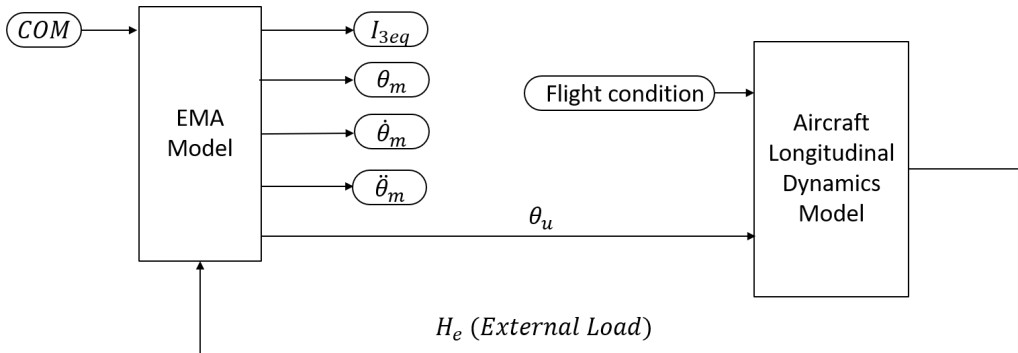

**Figure 2.** Overview of the top level of the Simulink model.

The *COM* block is used to generate an input for the system in terms of the desired deflection of the elevator of the aircraft over time. As one can observe, the system is controlled through the angular position of the control surface $\theta_u$, which is an output of the trapezoidal EMA block. The desired $\theta_u$ is specified at each timestep by the *COM* block, which can thus be used to produce any kind of elevator stick command.

The relevant logged signals are motor angular acceleration $\ddot{\theta}_m$, single-phase equivalent instantaneous current $I_{3eq}$ (which are then used to calculate the motor torque and the inertial torque of the motor) and the hinge moment $H_e$ (indicated as external load in the model) on the moving surface. These signals are then used to reconstruct TR and the residual torque, as explained in Section 2.6.

A macro approach was used to model the system for two reasons: On one hand, this makes for a computationally lighter model, and on the other hand it better suits the prognostic philosophy. The only appreciable advantage of modeling the system on a sub-component level for the purpose of fault identification is to be able to isolate a fault in a sub-component. In real-world applications of the form of this work, this would be useless unless there was a method to mechanically isolate or replace said sub-component without isolating or replacing the whole component. For instance, while it is certainly possible to model a ball-screw gear considering the variations in the friction of each single ball across the entire length of their canal and to identify a fault in a single or multiple ball, it would be impossible to isolate single balls in case of need, and the whole gear would be replaced anyway during ground maintenance since it is easier and safer.

Modeling each component using "mean" or "equivalent" properties better fits the modern principles of onboard safety and of ground maintenance; simultaneously, it is also faster, computationally lighter and more generalizable to model the system in this manner.

Figure 3 shows the model of the EMA [28], which will now be described in greater detail. It represents an electric BLDC motor connected to the elevator of an aircraft by a ball-screw reducer. The controller model is a simple proportional gain used to convert the error on the desired elevator position into a piloting current, with saturation blocks that

are used to keep the angular velocity of the motor and its current under their respective maximum values. In the case of the current, the saturation block acts as a over-current protection system.

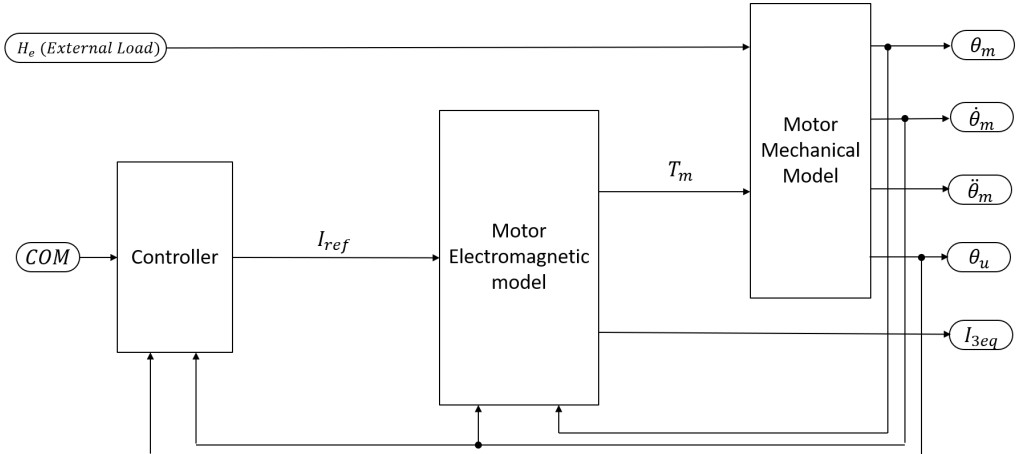

**Figure 3.** Overview of the top level of the EMA Mdel.

### 2.2. Trapezoidal BLDC Model

The simplified model of a trapezoidal BLDC used for this study is shown in Figure 4. As previously stated, it is derived from [27].

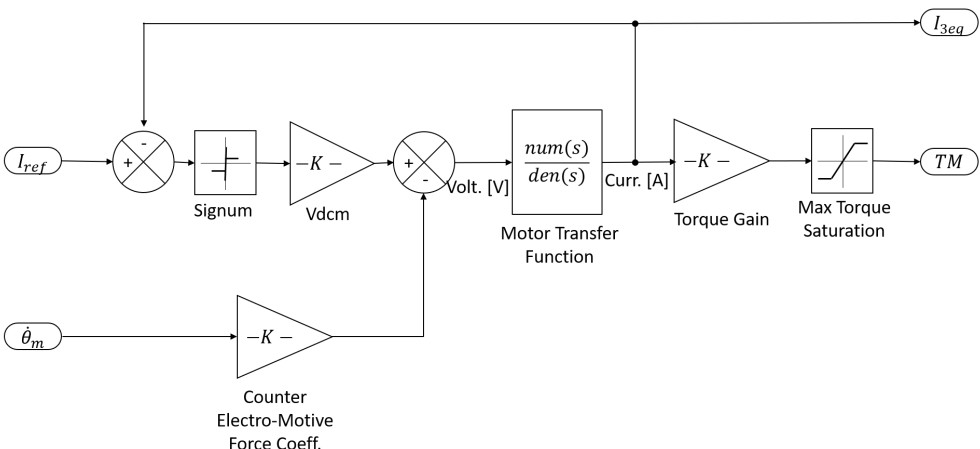

**Figure 4.** Motor Electromagnetic Model.

The model represents a 3-phases 4-poles BLDC trapezoidal motor. The BLDC motor is modeled as an equivalent single-phase current-piloted DC motor [7]. The error between the reference current $I_{ref}$ from the controller and the actual current in the stator coil $I_{3eq}$ is passed through a signum block, and the resulting signal is multiplied by the line voltage of the DC power supply in order to obtain the supplied voltage $V_{dcm}$. The interaction of the error with the signum block closely resembles a PWM control logic, where the frequency of the carrier wave is proportional to the integration timestep of the model.

The BEMF is then subtracted from the line voltage, and the motor transfer function is applied to obtain the stator coil current $I_{3eq}$ and the motor torque $TM$ by using the torque gain constant of the motor. A saturation block ensures that the motor does not supply a torque higher than its maximum torque.

The following equation shows the motor transfer function.

$$f(s) = \frac{1}{R + s \cdot L} \tag{1}$$

It represents a simple ohmic-inductive model of an electric motor.

The numerical value of the integration timestep of the model must be at least one order of magnitude lower than the smallest characteristic time of any phenomenon modeled. The integration timestep of this model has been set to $10^{-6}$ s since using a higher value would incur numerical limit cycles in the electrical model .

### 2.3. Mechanical Transmission Model

The mechanical transmission sub-model (shown in Figure 5) is a second order model with a single degree of freedom.

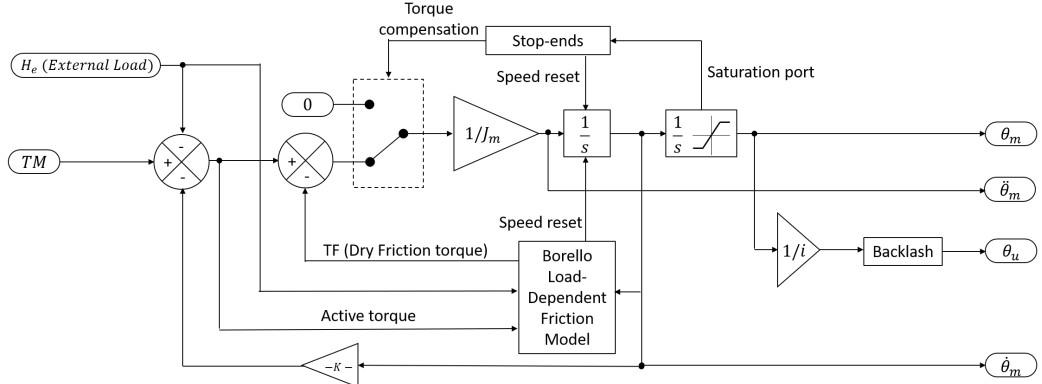

**Figure 5.** Motor mechanical model.

Viscous and inertial effects are considered, and non-linear effects such as backlash are considered too. Coulomb friction torque is evaluated through a load dependent Borello model with efficiency, described in the following section and more in depth in Section 2.4 and [29]. The angular acceleration of the motor's shaft $\ddot{\theta}_m$ is computed from its torque and the signal is then integrated two times in order to obtain the motor's shaft angular speed $\dot{\theta}_m$ and its angular position $\theta_m$. Speeds and positions are saturated as needed according to the maximum possible speed and the mechanical stop-ends. Finally, the angular position of the user's shaft $\theta_u$ is evaluated by applying the gear ratio to the angular position of the motor's shaft.

### 2.4. Friction Model

Friction fault modes can be described as particular kinds of mechanical and structural fault modes caused by the effect of an increment (or decrement) of the friction between two or more components. Friction causes mechanical wearing, which in turn is responsible for the deterioration of mechanical parts and their surfaces and, eventually, the production of metal or plastic flakes. As friction in the system increases, the motor requires more current to sustain the same level of output torque, reducing the overall power efficiency of the system or even damaging electrical and electronic components due to overcurrent or excessive heat. Excessive friction can eventually jam the actuator, resulting in possibly catastrophic consequences.

In this work, friction was modeled by using a set of parameters, i.e., static and dynamic friction torques and transmission efficiency under opposing and aiding loads. Deviations from nominal values indicate performance degradation in the actuator, and thus estimating their values is the goal of this work.

The friction model employed in this work is based on Borello's Friction Model [29], which is itself an evolution of Coulomb's dry friction model [30]. It is able to model both load dependent and load invariant friction torques. According to Coulomb's hypothesis, the load invariant friction torque $T_F$ is equal to its static (sticking) value $FSJ$ if the driving torque $T_M$ is not greater than $FSJ$ itself, and it is equal to its dynamic (slipping) value $FDJ$ if the contrary case is true. In the former case, the model must not set the body in motion $T_M < FSJ$, as it would be a violation of Newton's Third Law of Motion; in the latter case,

the body is set in motion, and the model must take into account whether the angular speed $\dot{\theta}$ is null or not. This translates into the following system of equations:

$$T_F = \begin{cases} T_M, & \dot{x} = 0 \wedge |FM| \leq FSJ \\ FDJ \cdot sgn(FM), & \dot{x} = 0 \wedge |FM| > FSJ \\ FDJ \cdot sgn(\dot{x}), & \dot{x} \neq 0 \end{cases} \tag{2}$$

Following the derivation of the load dependent model in [29], the load dependent friction torques are then accounted for by considering the total friction torque $T_F$ as the sum of the aforementioned load invariant part ($FSJ$ or $FDJ$) and a load dependent part. The load dependent friction torque is proportional to the load $F_R$ through the means of the efficiency of the transmission, but it must account for the verse of the load compared to the driving force. When the load acts in the same direction of the motion, the load is *aiding* the movement, while, on the contrary, the load is *opposing* the movement when it acts in the opposite direction of the motion. The opposing efficiency $\eta_O$ and the aiding efficiency $\eta_A$ of the transmission are defined, respectively, as the ratio between resisting and driving power and its inverse. The total friction torque in dynamic conditions is then calculated as follows.

$$T_F = \begin{cases} FDJ + (1 - \eta_A) \cdot |F_R|, & \text{under Aiding load} \\ FDJ + (\frac{1}{\eta_O} - 1) \cdot |F_R|, & \text{under Opposing load} \end{cases} \tag{3}$$

The total friction torque in static conditions is calculated by introducing the static-to-dynamic friction ratio $FSD$ and multiplying the dynamic value by $FSD$.

What is noteworthy is that the value of $\eta_O$ must be between 0 and 1. Negative values would mean that the friction aids the motion. The value of $\eta_A$ could be negative but must always be less than 1, and it is actually a measure of the irreversibility of the mechanical system. A value of $\eta_A$ between 0 and 1 means that the transmission is reversible. For $\eta_A = 0$, the friction torque is equal and opposed to the load, and their combined effect is null. Finally, for $\eta_A < 0$ the system is irreversible, and the resulting friction opposes the action of the motor.

Furthermore, a relation between $\eta_A$ and $\eta_O$ can be established through the gear ratio $\tau$ [31] as follows.

$$\eta_A = \frac{2\tau^2\eta_O - \tau^2 + \tau}{\tau^2\eta_O + \tau - \eta_O + 1} \tag{4}$$

### 2.5. Aircraft Longitudinal Dynamics Model

The model of longitudinal dynamics of the aicraft used in this work is taken from [32] and is shown in Figure 6.

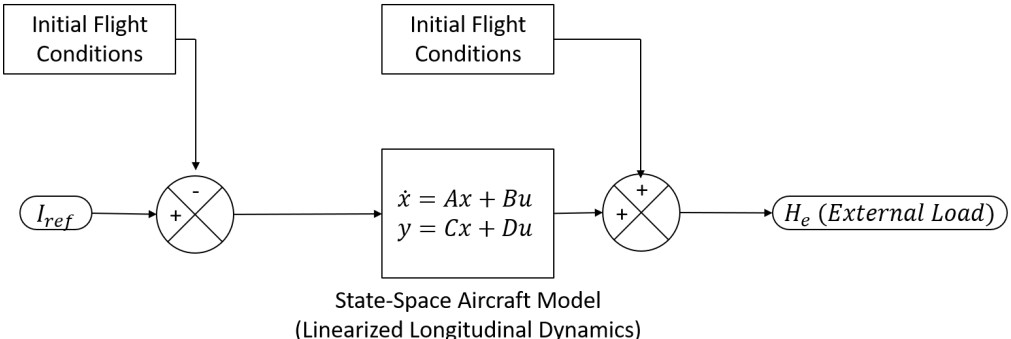

State-Space Aircraft Model
(Linearized Longitudinal Dynamics)

**Figure 6.** Aircraft Longitudinal Dynamics Model [32]. Airspeed, angle of attack, rate of pitch and other variables related to the dynamic behavior of the aircraft are also calculated but are not shown for the sake of clarity.

It is the means through which the actual angular position of the control surface is calculated (an elevator is used). It is a state space model of the linearized longitudinal dynamics of an aircraft.

*2.6. Residual Torque Reconstruction*

The inputs used by the neural network to estimate the targets must contain information about the targets themselves. The neural network tries to find a correlation between the inputs and the targets through regression; thus, if no such correlation exists or is shown to be weak, the predictions of the neural network will be inaccurate or even totally random.

For these reasons, in this work, the main input variable (and the main output signal of the Simulink model) is the residual torque defined as the sum of all friction torques in the transmission of the EMA.

The residual torque contains in itself information about the friction. Should friction unexpectedly increase or decrease, the residual torque would change accordingly. Equation (3) is hereby reported in its full formulation (for both static and dynamic conditions) for the sake of convenience.

$$
T_F = \begin{cases}
FSJ + (1 - \eta_A) \cdot |F_R|, & \text{Static conditions, under Aiding load} \\
FSJ + (\frac{1}{\eta_O} - 1) \cdot |F_R|, & \text{Static conditions, under Opposing load} \\
FDJ + (1 - \eta_A) \cdot |F_R|, & \text{Dynamic conditions, under Aiding load} \\
FDJ + (\frac{1}{\eta_O} - 1) \cdot |F_R|, & \text{Dynamic conditions, under Opposing load}
\end{cases}
\tag{5}
$$

It is evident how $\eta_A$, $\eta_O$, $FSJ$, $FDJ$ and indirectly $FSD$ are related to the residual torque through the dry friction torque. Viscous friction torque was neglected as its magnitude is usually negligible in aerospace EMAs. This fact can be attributed to the absence of operating fluids in EMAs when compared to other types of actuators, e.g., hydraulics-based actuators. Rotor ventilating effects, which are present, are usually very small when compared to motor torque.

It is now clear that the residual torque may be a good prognostic indicator as it holds information about the friction torques and efficiencies of the transmission. These values are direct indicators of mechanical wearing and degradation of the transmission.

Directly solving Equation (5) is not practical since it implies the knowledge of the parameters the algorithm tries to estimate.

However, the following relationship applies to EMAs (and, in general, mechanical transmissions connected to a mechanical user through a reducer):

$$
T_M = J_M \ddot{\theta}_m + F_{V_m} \dot{\theta}_m + H_e + T_R
\tag{6}
$$

where $T_M$ is the motor torque (the $m$ subscript means that the values are reduced to the motor shaft (fast shaft)), $J_m$ is the moment of inertia of the user (reduced to the fast shaft through the gear ratio squared), $\ddot{\theta}_m$ denotes the acceleration of the fast shaft, $\dot{\theta}_m$ is its speed, $F_{V_m}$ denotes the viscous friction coefficient, $H_e$ is the external load (hinge moment on the elevator) and $T_R$ is the residual torque. The first term on the right side of the equation is the inertial torque of the elevator.

As already discussed, the viscous friction torque can be neglected. The motor torque can also be expressed as $T_M = k_{CEMF} I$, where $k_{CEMF}$ is the motor torque coefficient and I the stator current. In light of this, by rearranging Equation (6), one can obtain the following.

$$
T_R = k_{CEMF} I - J_M \ddot{\theta}_m - H_e
\tag{7}
$$

The residual torque is equal to the motor torque minus the inertial torque and the external load. The constants ($k_{CEMF}$ and $J_m$) are known values once the system is characterized. The current I is already measured by the system itself for other purposes (such as piloting the BLDC motor), and the shaft acceleration $\ddot{\theta}_m$ is known once the shaft's speed $\dot{\theta}_m$ is known; it is measured by the hall sensors of the BLDC motor. The external moment

(hinge load on the elevator) can either be directly measured with pressure sensors on the elevator or be measured by the flight computer or be calculated once the state matrix of the aircraft and its controls are known (i.e., once the aircraft is characterized). In short, the signal of the residual torque can be reconstructed by using other already measured signals and some known constants unrelated to the mechanical faults.

The previous equations also show that the friction torques and the transmission efficiencies are indirectly dependent on other parameters, that could improve the accuracy of the network should the sole residual torque prove insufficient. In particular, the decision was made to also investigate the combination of residual torque $T_R$, fast shaft speed $\dot{\theta}_m$ and external load $H_e$ as to whether or not the speed being null determines if the system is in static or dynamic friction conditions, and a combination of the signs of speed and external load determines whether the system is working under aiding or opposing load. Figure 7 shows the trend of the residual torque over time during the actuation of the elevator, while Figure 8 shows the trend of $T_R$, $H_e$ and $\dot{\theta}_m$ as a 3D graph.

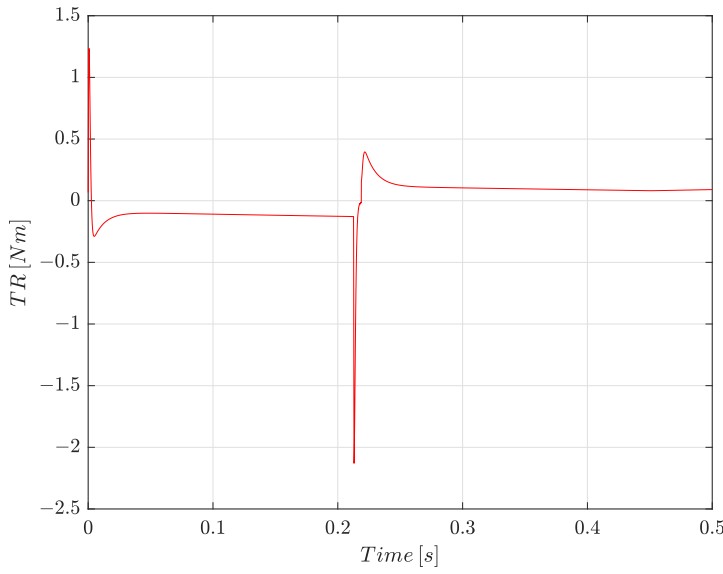

**Figure 7.** Residual torque trend over time during a simulation. The sudden spike at about $0.21s$ is due to the fact that the motor starts rotating in the opposite direction at that point.

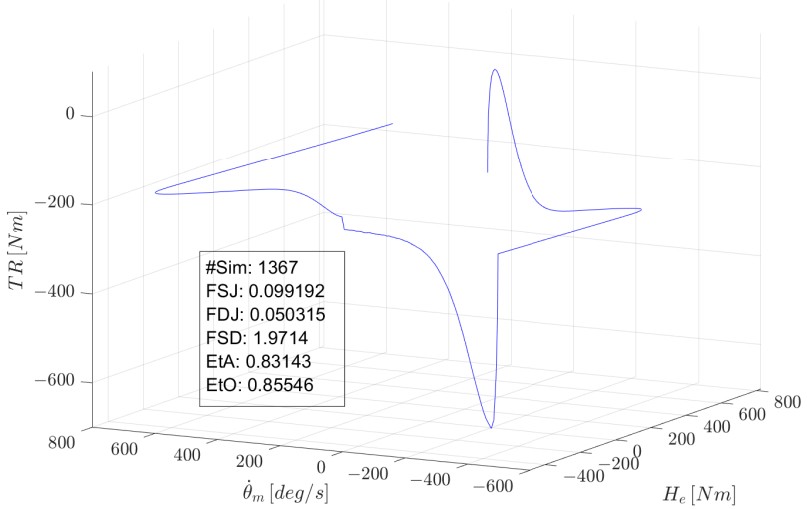

**Figure 8.** Trends of $T_R$, $H_e$ and $\dot{\theta}_m$ shown in a 3D space. The box shows the ID of the simulation this graph refers to and the values of the parameters used for the simulation.

### 2.7. Dataset Generation

The Simulink model used in this work has already been described in detail in Sections 2.1–2.5. With respect to the simulation themselves, each one is by performed by using a different random combination of possible values for the parameters of interest ($\eta_A$, $\eta_O$, $FSJ$ and $FDJ$). Since, in prognostics, it is important to derive the trend of the health of the system from small variations of its parameters (otherwise the problem falls into the domain of diagnostics), only small deviations from the nominal values are considered, and thus the possible values for each of the parameters of interest are in a ±20% range of their nominal value. The values of $\eta_O$ are calculated from those of $\eta_A$ by using Equation (4). It is assumed that such parameters are constant during the simulation as in real life since the total simulation time is orders of magnitude shorter than the timescale at which the degradation of performance due to friction happens.

The command given to the system specifies the desired elevator position over time and is a third order polynomial defined as in Table 1 and shown in Figure 9 unless otherwise specified in the results. Since any command can be used in the framework of this analysis, a polynomial command curve has been chosen. The choice has been made considering that it provides a good compromise between ease of implementation and numerical stability of the model, while still providing potential for further generalization.

**Table 1.** Points used to define the third order polynomial command curve.

| Point # | Time (s) | Position (°) |
|---------|----------|--------------|
| 1 | 0 | 0 |
| 2 | 0.1667 | −15 |
| 3 | 0.3333 | 0 |
| 4 | 0.5 | 15 |

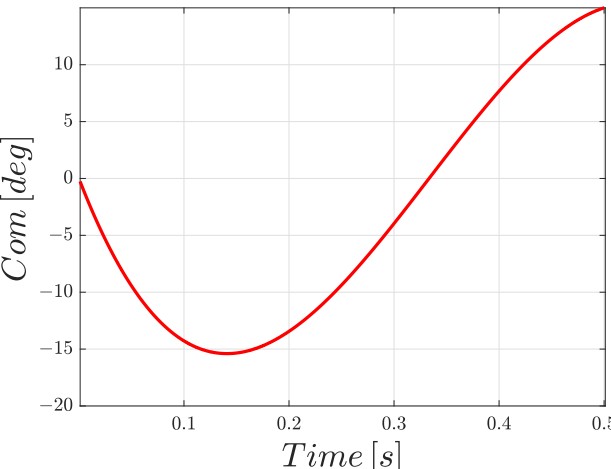

**Figure 9.** Elevator command curve used for the simulations.

The model has a simulation timestep of $10^{-6}$ s, meaning that $5 \times 10^5$ values for each variable are calculated and logged during each 0.5 s simulation. Each dataset is composed of 2000 simulations, but, as will be discussed in the following sections, a smaller number of simulations and timesteps can be extracted from the dataset during data pre-processing for the ANN.

As it is relevant, the difference in datasets across the two cases examined. The difference is in the model itself: Case B simulations employ a full model of the longitudinal dynamics of the aircraft, while Case A simulations are performed on the same system but stripped of the aircraft dynamics: instead, the aircraft longitudinal dynamics model, which is used to calculate the hinge moment, is substituted by a constant block, which simulates the torque applied on the control surface by its own weight.

### 2.8. Artificial Neural Networks

The use of artificial neural networks (ANNs) for regression in complex problems is becoming more and more common since they offer the flexibility that "classical" regression algorithms lack. ANNs are capable of finding correlations between their inputs and outputs even when the best regression equations for the data are not known or when it is not yet known if the data are correlated at all [33].

In line with other studies, such as the already referenced [25], shallow artificial neural networks in feed-forward configuration were employed for this work.

A very simple neural network is used for the complex task of multi-input multi-output regression, with good effects; it is, however, possible that other statistical and machine learning methods, of similar complexity, could perform as well as a feed-forward neural network.

As previously mentioned, $\eta_A$, $\eta_O$, $FSJ$ and $FDJ$ are the outputs of the ANN, while the residual torque $T_R$ is sampled and used as input for the ANN.

A decision was made to program and train the ANN by using MATLAB and its Deep Learning Toolbox, as the software offers a variety of options at a reasonable performance level, and it simplifies the passage of data from Simulink simulations to the ANN.

The generic final architecture for this application is presented in Figure 10. Since the inputs are technically a time sequence while the outputs are time invariant, a decision was made to use a number of inputs nodes equal to the number of timestep of the sequence. However, in order to reduce complexity and training times, the number of selected timesteps was reduced through interpolation. For instance, due to the integration timestep of the model, 0.5 s of simulation equated to 50,000 values of a variable, one every $10^{-6}$ s. Reducing the number to 2000 means that the signal is interpolated so that, now, each apparent timestep spans $2.5 \times 10^{-4}$ s.

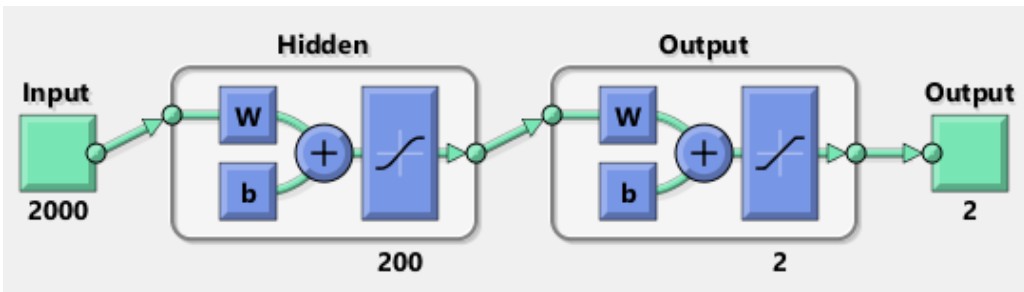

**Figure 10.** General ANN architecture used in this work.

The size of the hidden layer varies according to the study case, as it is one of the first things to modify when seeking better performances from the network. A simple algorithm was implemented to exclude bad data (numerical limit cycles, numerical divergence, non-physical scenarios, etc.) from the training dataset. It is then easy to implement another algorithm that converts data of the real system from the known format to the specific sampling requirement of the chosen neural network through linear interpolation of the logged values of each input variable.

As previously stated, $\eta_A$ and $\eta_O$ are linked by Equation (4). It was also assumed that the gear ratio of the transmission remains constant during the life cycle of the transmission itself, as it is rare that a change in the value of the gear ratio is not a result of a fault that completely jams the actuator. For these reasons, it was chosen to reduce the complexity of the ANN by predicting only the values of $\eta_A$, $FSJ$ and $FDJ$.

With respect to the specifics of the training of the ANN, Scaled Conjugate Gradient Backpropagation (MATLAB's *'trainscg'* function) was used as the training function since it showed the best compromise between training time and accuracy. Inputs (residual torque) and outputs (friction parameters) are normalized in a [0, 1] range (meaning that the nominal value for each friction parameter is 0.5) as it is good practice in this sort of problems, and it renders it easier to interpret the results. Linear Saturation (MATLAB's

'*satlins*' function) is applied as the transfer function between the hidden and output layer of the network so that predicted values outside the [0–1] range are brought to the closest boundary of the range. Before the training starts, data are randomly divided into training data (75% of the simulations), validation data (15% of the simulations) and test data (10% of the simulations). Finally, mean squared error is used as the performance metric (with a target value of $10^{-6}$, chosen arbitrarily), and the training stops if the performance of the network on the validation portion of the data does not improve for 12 consecutive iterations.

From now on, the number of simulations used out of each dataset for each training will be referred to as "Number of samples", while the number of linearly interpolated timesteps of the reconstruction of the residual torque will be referred to as "Number of inputs (of the ANN)" or simply as "Number of timesteps".

## 3. Results

### 3.1. General Results and Considerations

Trial and error immediately showed that the trained networks were unable to correctly predict the values of $FSJ$ under any circumstances, and in fact the inclusion of these parameters among the outputs decreased the general performance of the networks. This is because the actuator works under stick conditions only at the start and when the direction of motion changes, as shown in Figure 11; it is only during those instances that its angular speed is null, and the motor torque has yet to overcome the static friction. Even when using commands that causes the angular speed to have multiple zero-crossings (as a sinusoidal command, for instance), the issues was not solved. Suggestions to overcome this problem are discussed in the next chapter, but for now let it be known that networks for Case A and Case B will not include $FSJ$ among their outputs; thus, the networks will only predict the values of $FDJ$ and $\eta_O$ and, by extension, $\eta_A$.

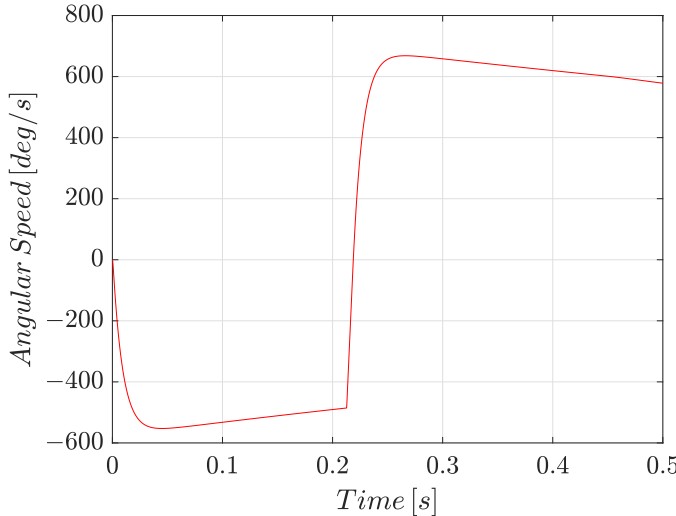

**Figure 11.** Graph showing that the system works under stick conditions only two times during the whole simulation.

### 3.2. Case A: Same Command; No Aerodynamic Loads

This case can be thought of as an aircraft that, while on ground, performs a specific command sequence before or after a flight or during maintenance events to check the health status of some of its components.

Even though the trend of the residual torque over time is similar to previous tests, including the aircraft's dynamics block, the constant external load seems to lose information regarding the efficiency of the transmission that should be contained, at least in principle, in the residual torque.

Recalling Equation (5), as previously mentioned, this behavior is not completely surprising since external load multiplies the efficiency both under aiding and opposing conditions. Even in adding the angular speed of the fast shaft to the input vector for the ANN, the performance did not change noticeably.

There was no improvement on prediction accuracy by using $\eta_A$ in place of $\eta_O$ as the target, meaning that this behavior is not dependent on whether the actuator is operating under aiding or opposing load conditions. However, the network is capable of accurately predicting $FDJ$.

The results shown in Tables 2 and 3 and Figure 12 were obtained after $FDJ$ was left as the only output for the network. A network architecture with 100 neurons, 1000 samples and 2000 timesteps was used since it obtained the most accurate prediction out of all the attempted setups .

**Table 2.** Performances for Case A.

| Type of Performance | MSE |
|---|---|
| Training | $2.417 \times 10^{-4}$ |
| Validation | $2.433 \times 10^{-4}$ |
| Test | $2.453 \times 10^{-4}$ |

**Table 3.** Validation for Case A. Ten random predicted values were confronted with their actual value ("Target value") in order to calculate the relative error of the prediction.

| Parameter: $FDJ$ | | |
|---|---|---|
| **Target Value** | **Predicted Value** | **Relative Error (%)** |
| 0.9782 | 0.9792 | 0.11 |
| 0.8399 | 0.8442 | 0.50 |
| 0.2670 | 0.2722 | 1.98 |
| 0.2687 | 0.2614 | 2.73 |
| 0.5300 | 0.5160 | 2.64 |
| 0.6933 | 0.7047 | 1.64 |
| 0.4183 | 0.4505 | 7.69 |
| 0.6042 | 0.6013 | 0.49 |
| 0.6015 | 0.5843 | 2.85 |
| 0.8478 | 0.8707 | 2.71 |

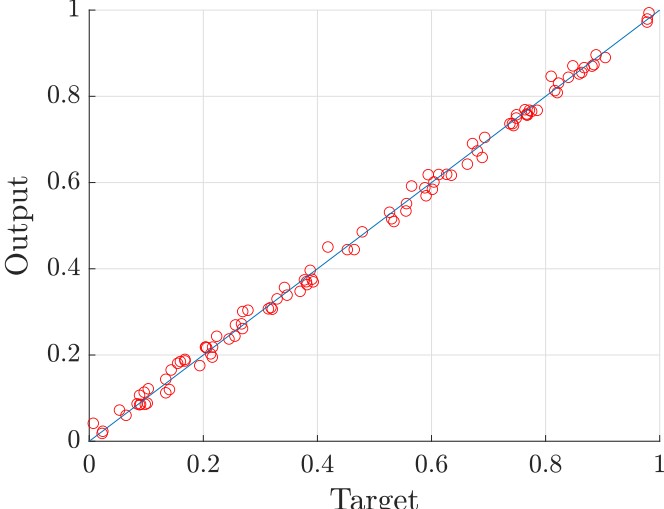

**Figure 12.** Fit for FDJ of Case A. The values on the x-axis are the actual values of the parameters (normalized between 0 and 1), while the values on the y-axis are those predicted by the network.

Further discussions of the results of this case are reported in the next section.

### 3.3. Case B: Same Command; with Aerodynamic Loads

The importance of this case is that it can represent an actuation during flight conditions, with the only limitation posed by a constant actuation command. However, the actuation command used to train the network can be tailored to represent the actual actuation sequence observed during flight conditions.

The combinations of neurons number, samples and timesteps were analyzed, and the results are reported in Table 4. The results reported in Tables 5 and 6 and Figure 13 refer to the network with 50 neurons and 500 timesteps (or inputs) trained on 500 samples. Based on the results reported in Table 4, these settings seem to be the point where adding complexity to the network and/or the training hits the point of diminishing returns, and the training time rises exponentially with no substantial meaningful performance gain. All the training stopped because of the validation condition (i.e., the validation MSE failed to improve for 12 consecutive epochs).

Interestingly, the predictions of $\eta_O$ have, on average, a smaller relative error than those of $FDJ$, and the predictions in general are worse the more the target value becomes closer to the boundaries of the range of the parameter.

**Table 4.** Settings tried for Case B and relative performances.

| Timesteps | Simulations | Neurons | Training MSE | Test MSE |
|---|---|---|---|---|
| 100 | 100 | 10 | $6.007 \times 10^{-6}$ | $1.611 \times 10^{-5}$ |
| 500 | 100 | 50 | $4.364 \times 10^{-5}$ | $4.244 \times 10^{-5}$ |
| 100 | 500 | 10 | $6.217 \times 10^{-6}$ | $6.730 \times 10^{-6}$ |
| 100 | 500 | 50 | $4.529 \times 10^{-6}$ | $7.643 \times 10^{-6}$ |
| 500 | 500 | 50 | $1.742 \times 10^{-6}$ | $2.601 \times 10^{-6}$ |

**Table 5.** Performances for Case B.

| Type of Performance | MSE |
|---|---|
| Training | $1.742 \times 10^{-6}$ |
| Validation | $1.926 \times 10^{-6}$ |
| Test | $2.601 \times 10^{-6}$ |

**Table 6.** Validation for Case B. Ten randomly predicted values were confronted with their actual value ("Target value") in order to calculate the relative error of the prediction.

| Parameter: $FDJ$ | | | Parameter: $\eta_O$ | | |
|---|---|---|---|---|---|
| Target Value | Predicted Value | Relative Error (%) | Target Value | Predicted Value | Relative Error (%) |
| 0.9648 | 0.9596 | 0.54 | 0.0761 | 0.0756 | 0.73 |
| 0.5432 | 0.5415 | 0.30 | 0.0734 | 0.0730 | 0.45 |
| 0.3115 | 0.3107 | 0.26 | 0.6876 | 0.6871 | 0.08 |
| 0.4217 | 0.4195 | 0.51 | 0.0984 | 0.0979 | 0.57 |
| 0.5003 | 0.4987 | 0.32 | 0.5235 | 0.5224 | 0.22 |
| 0.5839 | 0.5842 | 0.06 | 0.6001 | 0.6012 | 0.19 |
| 0.2609 | 0.2608 | 0.02 | 0.8887 | 0.8914 | 0.31 |
| 0.2121 | 0.2123 | 0.12 | 0.7008 | 0.7015 | 0.10 |
| 0.9311 | 0.9312 | 0.02 | 0.1336 | 0.1349 | 0.96 |
| 0.7147 | 0.7158 | 0.16 | 0.6471 | 0.6499 | 0.43 |

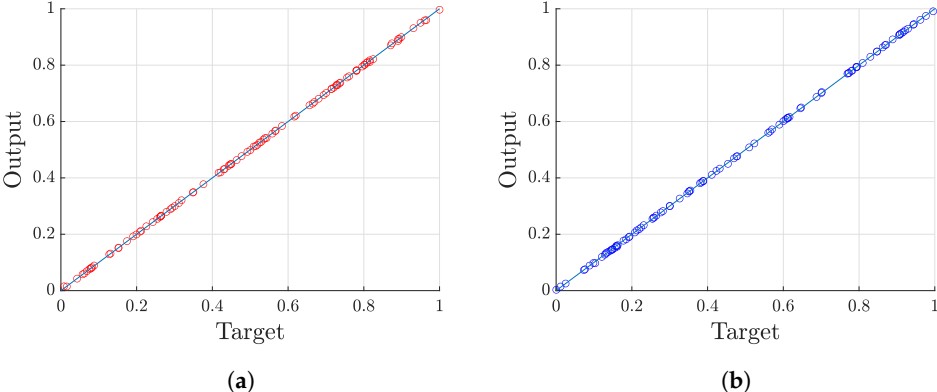

(**a**)　　　　　　　　　　　　　(**b**)

**Figure 13.** The values on the x-axes are the actual values of the parameters (normalized between 0 and 1), while the values on the y-axes are those predicted by the network. This figure shows the individual fits for the following: (**a**) $FDJ$ of Case B; (**b**) $\eta_O$ of Case B.

### 3.4. Extended Case A

The analysis for this case was conducted by using the same principles as Case A but with a ramping external load instead of a constant one.

This case was tested to observe how the network prediction is changed when using a non-constant external load; in this sense, case A extended is more of a synthetic case with no immediate and direct real-life analogy.

The parameter $\eta_O$ was also reintroduced among the outputs of the neural network. The findings of this case, reported in Table 7 and Figure 14, will be discussed in next section. The same 100 neurons, 1000 samples and 2000 timesteps network architecture as in Case A were used.

**Table 7.** Performances for Extended Case A.

| Type of Performance | MSE |
| --- | --- |
| Training | $4.586 \times 10^{-4}$ |
| Validation | $5.320 \times 10^{-4}$ |
| Test | $5.542 \times 10^{-4}$ |

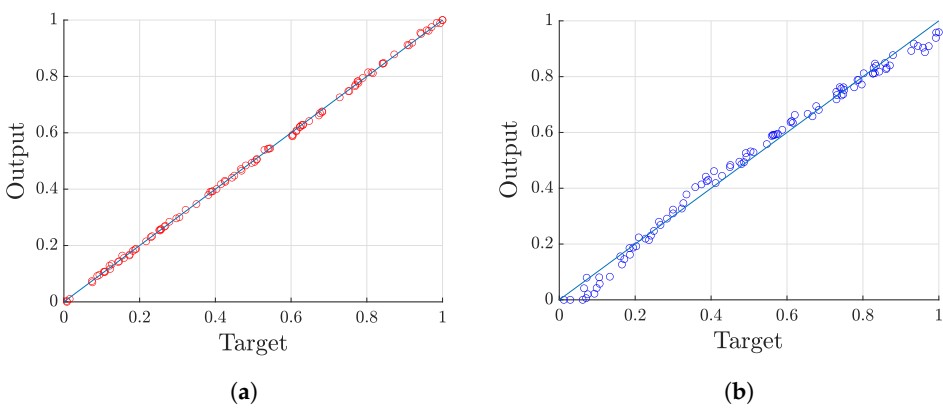

(**a**)　　　　　　　　　　　　　(**b**)

**Figure 14.** The values on the x-axes are the actual values of the parameters (normalized between 0 and 1), while the values on the y-axes are those predicted by the network. This figure shows the individual fits for the following cases: (**a**) $FDJ$ of Extended Case A; (**b**) $\eta_O$ of Extended Case A.

### 3.5. Generalization of the Results for Different Command Curves

One of the goals of the analysis was to determine whether or not the results are dependent on the command curve used to perform the simulation. Since it is the command curve itself that determines the response of the system, the command curve indirectly influences the residual torque. It is also certainly true that the system response logged or

simulated with a command would produce unusable and incoherent results when used as input for an ANN trained on a different command, but our thesis was that ANNs trained on different commands should roughly have the same performances. This thesis was proven by repeating Case B with different datasets. For instance, performances reported in Table 8 refer to a network trained on a dataset generated by using the command curve shown in Figure 15. A number of these experiments was performed, proving that ANNs can be successfully trained to the same extent regardless of the command curve used.

**Table 8.** Performances for Case B performed by using a different command curve (Figure 15).

| Type of Performance | MSE |
|---|---|
| Training | $3.349 \times 10^{-6}$ |
| Validation | $3.296 \times 10^{-6}$ |
| Test | $2.344 \times 10^{-6}$ |

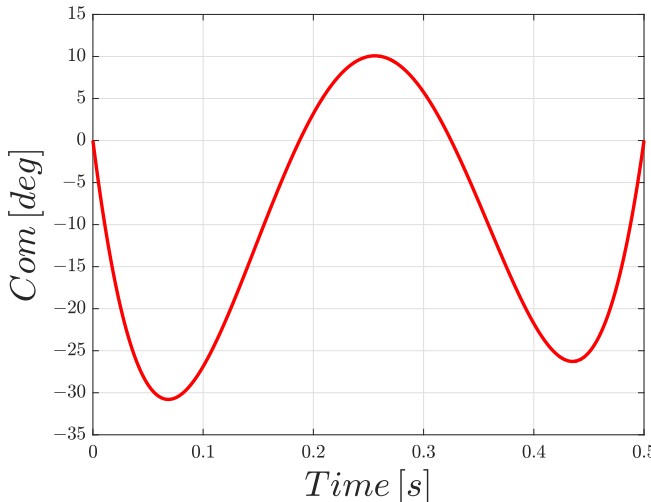

**Figure 15.** One of elevator command curves used when testing the generalization of Case B.

## 4. Discussion

The results achieved, especially for Case B, show the great potential of the proposed methodology. However, improvements can be made not only for tailoring and optimizing the process for real-life applications but also for improving its weaknesses.

For instance, the values of FSJ may be predicted by using specifically engineered elevator command sequences that exploit the dynamics of the system in order to make it continuously operate in stick-slip conditions; that is definitely a point to be considered in further studies on the subject.

With respect to Case A and Case B, relative errors of under 1% were achieved for values close to the nominal values of the parameters. Larger errors close to the boundary values of the variation range are actually not worrisome, as deviations of 10–15% from the nominal values of the efficiency of the transmission or the friction torques already fall into the domain of diagnostics; there would be little utility in the prediction a value that is far off for prognostic purposes, as such a deviation would be clearly noticeable in how the system responds to a command.

Aside from the performance of the networks, another important result to comment on is the fact that networks seem to be unable to predict both efficiencies when the external load is constant. A qualitative explanation of this behavior is related to the nature of the neural networks itself: the network simply is not exposed to enough variety during training, especially when the external load is constant or worse, null.

For the sake of thoroughness, a case we called "Extended Case A" was also analysed. The resulting MSE in the order of $10^{-4}$ shows that the presence of even a simple non-

constant external load makes the difference in the ability of the networks to correctly predict the efficiency parameters, meaning that Case A could be put into practice by using some kind of system to apply a predictable external load on the control surface while it is actuated on ground for data logging.

As for the creation of a dataset from physical systems, the question of collecting real, labeled data arises. The main idea is to create a digital twin of a servoactuation system with experimental validation and then to modify the model parameters in order to simulate faults of desired value.

## 5. Conclusions

In this paper, a new methodology for the estimation of the friction torque of an EMA transmission has been proposed. A virtual sensor approach for the generation of a synthetic quantity, the residual torque, has been adopted. Residual torque is then sampled and used as input in a neural network that can directly predict, with good accuracy, both the dynamic friction torque (FDJ) and the opposing transmission efficiency ($\eta_O$) and, indirectly, the aiding transmission efficiency ($\eta_A$).

The algorithm has been tested in order to evaluate the generalization potential, both relative to a generic actuator command and considering external loads, which appears satisfactory. However, the next step is to apply the proposed methodology in a generic in-flight condition during normal operations and to check whether the results prove to be conclusive or if the increase complexity renders the proposed method inapplicable.

In any case, the implementation of the algorithm during routine maintenance operations is simple and could be an effective tool in the evaluation of the actual health status of an EMA transmission.

Finally, some future developments will include the application of different types of neural networks, possibly evaluating the performance of Higher-Order Neural Networks and the robustness and comparability of the results obtained when using different friction models.

**Author Contributions:** Conceptualization, P.C.B. and M.D.L.D.V.; methodology, P.C.B. and G.Q.; software, A.I.; validation, G.Q. and A.I.; formal analysis, G.Q.; investigation, G.Q. and P.C.B.; resources, A.I.; data curation, A.I.; writing—original draft preparation, A.I.; writing—review and editing, G.Q.; visualization, G.Q. and A.I.; supervision, M.D.L.D.V. and P.M.; project administration, P.M.; funding acquisition, P.M. All authors have read and agreed to the published version of the manuscript.

**Funding:** This research received no external funding.

**Institutional Review Board Statement:** Not applicable.

**Informed Consent Statement:** Not applicable.

**Data Availability Statement:** The data presented in this study are openly available at FigShare at https://doi.org/10.6084/m9.figshare.14346419 (accessed on 7 August 2021).

**Conflicts of Interest:** The authors declare no conflicts of interest.

## Abbreviations

The following abbreviations are used in this manuscript:

| | |
|---|---|
| EMA | Electro-Mechanical Actuator; |
| RUL | Remaining Useful Life; |
| UAV | Unmanned Aerial Vehicle; |
| FDI | Fault Detection and Identification; |
| FDJ | Friction Dynamic Torque; |
| FSD | Friction Static-to-Dynamic Ratio; |
| FSJ | Friction Static Torque; |
| PHM | Prognostics and Health Management; |
| ANN | Artificial Neural Network; |

| BEMF | Back Electro-Motive Force; |
|------|---------------------------|
| CEMF | Counter Electro-Motive Force; |
| BLDC | BrushLess Direct Current; |
| PWM  | Pulse With Modulation. |

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
