# Peer review of "A New Method for Friction Estimation in EMA Transmissions"

_actuators, doi:10.3390/act10080194_

Round 1

Reviewer 1 Report

In the present manuscript, the authors investigated the viability of the residual torque as a prognostic indicator for EMAs in a neural network-based methodology. It is a well-defined research problem and is written in a very systematic manner. However, a few technical things to be addressed in the revised version of the manuscript

  1. Kindly add the Summary or Conclusion part, which is missing here
  2. Figures numbers are random, especially after FIGURE 12, suddenly FIGURE 15 is introduced. Please kindly rearrange the figures.
  3. Physical explanation and discussion are missing for Fig.18 and Fig. 19.
  4. Please define all physical assumptions, which have been considered in the proposed analytical model.
  5. Kindly increase the figure font (especially axis level and tick level) of Figures 15 & 18.
  6. Please explain the sharp drop down of angular speed with time (~0.22 sec) under stick conditions.
  7. Please add these papers in refs.: a). https://doi.org/10.1016/j.cja.2016.07.006 b) https://doi.org/10.1016/j.ifacol.2015.09.612
  8. Please spilt the following sentences. In the present form, it is very difficult to understand the main message.
  9. “In this work…” (line # 6)
  10. “In this context …” (line # 84)
  11. “In line with general philosophy…” (line #163)
  12. “It is now clear that…” (line # 221)
  13. “While the importance…” (line # 404), and so on.

Author Response

  1. Kindly add the Summary or Conclusion part, which is missing here.

Conclusions section has been added as suggested.

  1. Figures numbers are random, especially after FIGURE 12, suddenly FIGURE 15 is introduced. Please kindly rearrange the figures.

The numbering of figures 12 and 13 (13 and 18 using wrong previous numbering) has been fixed and is now incremental.

  1. Physical explanation and discussion are missing for Fig.18 and Fig. 19.

Figure 14 (ex figure 18) is the same plot as fig. 13, but pertaining to Extended Case A; the discussion is found in Subsec. 3.4, which only highlights the differences for the sake of brevity. Figure 15 (ex figure 19) is one of the many command curves used to prove the generalization of the model. Description is found in Subsec. 3.5 .

  1. Please define all physical assumptions, which have been considered in the proposed analytical model.

There are two main physical assumptions made in the model: lumped-parameters and temperature invariance, i.e. no thermal flows modeling is made. Since the simulations run-time is ~1s, the temperature invariance is a reasonable assumption. Future works will consider also thermal flows in the model.

  1. Kindly increase the figure font (especially axis level and tick level) of Figures 15 & 18.

The font in figures 15 and 18 (old numbering) has been increased to improve readability.

  1. Please explain the sharp drop down of angular speed with time (~0.22 sec) under stick conditions.

At ~0.21 s the motor starts running in reverse, and thus the sudden spike of residual torque (stick condition) and thus of the inertial component () and current flow (motor torque). Figure 7 caption explains the phenomenon.

  1. Please add these papers in refs.: a). https://doi.org/10.1016/j.cja.2016.07.006 b) https://doi.org/10.1016/j.ifacol.2015.09.612

The suggested references have been added (refs. 16-17).

  1. Please spilt the following sentences. In the present form, it is very difficult to understand the main message:
  • “In this work…” (line # 6)
  • “In this context …” (line # 84)
  • “In line with general philosophy…” (line #163)
  • “It is now clear that…” (line # 221)
  • “While the importance…” (line # 404), and so on.

The indicated sentences have been rephrased to increase readability.

Reviewer 2 Report

Comment 1:

Some writing errors:

  1. “FDI strategy are classified into”: Are à is
  2. “for real time of offline FDI, …”: of à
  3. “algorithms to analyze of one or more signals outputted by the system …”: analyze à

Comment 2:

The fault management is of three phases: 

Fault detection, fault diagnosis, and fault commendation, see the literature “ https://www.sciencedirect.com/science/article/pii/S1071581903002210”. In that paper, the human factor is included, as the fault management may be performed by humans through an interface.

The basic idea is to build a standard reference of behaviour of the underlying system. The actual or measured or observed behaviour is compared with the reference to determine any wrong with the system. 

Detection: deviation of the actual behaviour and the reference. 

Diagnosis: cause for the fault. 

Compensation: solution to remove the fault.

The authors need to clarify which phases of fault management their paper may have made contribution. 

Comment 3:

The main idea in this paper is to develop a model for the behaviour of the system under diagnosis. There are three strategies of DW opine the model: principal, empirical and combined principal and empirical, see the literature “Thermal-error modeling for complex physical systems: the-state-of-arts review”. I wonder if there is any contribution with this paper on modelling against the literature in the area of modeling.

Comment 4: 

It seems that the authors take an empirical approach or machine learning approach. They used ANN. I wonder if the so-called high order NN may improve the performance. In general, I feel that building an Ann model is a routine work. 

Comment 5:

There are several types of friction models. The paper appears only consider the Coulomb model. Needs some explanation, as the literature shows otherwise – “Experimental comparison of five friction models on the same test-bed of the micro stick-slip motion system” (mechanical science).

Author Response

Comment 1

  1. “FDI strategy are classified into”: Are à is
  2. “for real time of offline FDI, …”: of à
  3. “algorithms to analyze of one or more signals outputted by the system …”: analyze à

Thank you very much for your suggestions. We fixed them.

Comment 2

The authors need to clarify which phases of fault management their paper may have made contribution. 

The paper is centered on the estimation of the friction forces present in an EMA.
In this sense, the paper pertains to the prognostics field of PHM (prognostics and health management, see Vachtsevanos et al., Intelligent fault diagnosis and prognosis for engineering systems, 2006), considering that the systems to which the algorithm can be applied are systems that present faults (for which no location assumption is made) that produce an increase in the friction on the transmission line. In other words, we have hypothesized only an increase in friction torque deriving from some unspecified fault present in the transmission line. 
However, the systems studied do not pertain to diagnostics since the operation of the systems is still satisfactory and does not limit the function imposed.

Comment 3

[…] I wonder if there is any contribution with this paper on modelling against the literature in the area of modeling.

The system is modeled using a physical-based approach with a high level of accuracy. The only main assumptions made are of lumped-parameters and no heat flows which is justified by the short run time of the model (~1s). As for novelty, this definition of residual torque is presented which, as far as the authors know, has not been defined before in the same manner.

Comment 4

They used ANN. I wonder if the so-called high order NN may improve the performance. In general, I feel that building an Ann model is a routine work. 

The authors thank the reviewer for the suggestion of using higher-order NN for this specific application case and will certainly investigate their performance in future works. A note on this subject has been added in the Conclusions section.

Comment 5

The paper appears only consider the Coulomb model. Needs some explanation, as the literature shows otherwise – “Experimental comparison of five friction models on the same test-bed of the micro stick-slip motion system” (mechanical science).

The model used (Borello friction model) is capable to accurately evaluate stick-slip, dynamic and static friction conditions, breakaway resolution with good accuracy, as in Pennestri, Ettore, et al. "Review and comparison of dry friction force models." Nonlinear dynamics 83.4 (2016): 1785-1801 or  L. Borello, M.D.L. Dalla Vedova – Dry Friction Discontinuous Computational Algorithms – International Journal of Engineering and Innovative Technology (IJEIT), Vol. 3, No. 8, February 2014, pp. 1-8, ISSN: 2277-3754.

For this particular application, the authors have not deemed necessary adopt other, more complex friction model which are more expensive to run.

Reviewer 3 Report

Comments on the paper proposed by Quattrocchi et al: “A new method for friction estimation in EMA transmissions”.

Here are reported some considerations:

The proposed article concerns an interesting predictive approach for diagnosing aircraft failures using neural networks but presents some critical issues.

The title of the paper refers to a method for estimating the coefficient of friction, but the connection between the coefficient of friction and damage is not clear. Many fundamental definitions are not supported by scientific literature, for example the paragraph "2.7 Residual torque reconstruction", in which a questionable definition of residual torque is given without bibliographic references. Likewise, viscous torque is defined as negligible without scientifically justifying the reason.

The "materials and methods" part is too dispersive as it contains ten paragraphs, some of which are reported in the wrong section ( 2.1; 2.8), for example "scope of work" must be integrated in the introduction.

Ultimately it is clear that an increase in Friction Torque is correlated with an increase in CoF, but it is not clear how the CoF is estimated and especially where (as mechanical coupling) an increase in CoF is estimated.

It is also necessary to provide a nomenclature at the beginning of the paper.

For all the previous reasons, the reviewer recommends minor amendments of paper for publication in Actuators.

Author Response

Dear Reviewer,
First of all, we would thank you warmly for the quick response and your valuable suggestions. 

  1. The title of the paper refers to a method for estimating the coefficient of friction, but the connection between the coefficient of friction and damage is not clear. Many fundamental definitions are not supported by scientific literature, for example the paragraph "2.7 Residual torque reconstruction", in which a questionable definition of residual torque is given without bibliographic references. Likewise, viscous torque is defined as negligible without scientifically justifying the reason.

The definition of residual torque presented in the paper is probably novel, i.e. to the best of our knowledge there is not any other literature mention of ‘residual torque’ in the same context or meaning. However, the definition presented is very simple and arises from a simple dynamical equilibrium equation of the mechanical system. The residual torque includes all the loss terms. The other assumption made is that the viscous friction is negligible: this assumption is made considering an EMA in close-to-nominal conditions, thus all the elements that could generate viscous friction are engineered in order to minimize it as much as possible. Furthermore, given the almost absence of fluids (in comparison to, for example, an electrohydraulic actuator), the only elements providing viscous friction are the lubricated bearings and ventilation effect of the rotor, which are of small entity when compared to motor torque. A clarification on the matter has been added in Subsection 2.6 .

  1. The "materials and methods" part is too dispersive as it contains ten paragraphs, some of which are reported in the wrong section ( 2.1; 2.8), for example "scope of work" must be integrated in the introduction.

Subsection ‘Scope of the work’ has been moved to Section 1 ‘Introduction’. Subsection 2.8 ‘Application of interest’ has been removed since it mostly presented duplicated information already present in other sections.

  1. Ultimately it is clear that an increase in Friction Torque is correlated with an increase in CoF, but it is not clear how the CoF is estimated and especially where (as mechanical coupling) an increase in CoF is estimated.

We have noticed that in several locations the term ‘coefficient of friction’ was present. We would like to apologize for any confusion that might have arisen from using this wrong wording. We have corrected it using ‘friction torque’. In fact, the friction modeling used in the paper does not include any form of coefficient, rather it directly models the friction loss as a torque. As for the location of the increase of friction, it is not explicitly modeled in the paper. The algorithm is a distributed algorithm, which only evaluates increases (or rather, variations) of the friction torques along the mechanical transmission. This makes it possible to apply it, after proper parameters characterization, to any EMA which might have a different mechanical transmission to the one considered in this paper.

  1. It is also necessary to provide a nomenclature at the beginning of the paper.

Does the reviewer refer the Abbreviations list? If that is the case, it has been added at the end of the document as per template specification. More entries (such as FSJ, FDJ, etc.) have been added.

We hope to have properly addressed every point made.

Kind regards,

The authors

Round 2

Reviewer 2 Report

The paper reads like a routine exercise of ANN for parameter identification.

Author Response

Comments and Suggestions for Authors: The paper reads like a routine exercise of ANN for parameter identification.   The novelty of the paper is the definition of the residual torque as presented and the investigation on the viability of said quantity as a prognostic indicator. The neural network has been used in a classical sense to perform a regression analysis on the residual torque curve (including motor angular speed and external load) to evaluate the actual friction torque and the opposing efficiency. However, other machine learning methods could be applied instead of the neural network to achieve the same objective, i.e. estimation of the parameters of interest. Therefore, the authors chose to use a feedforward network given the nature of the problem. In fact, the problem is a multi-input multi-output regression task, and thus a simple feedforward neural network has been found to be very apt to solve the task with acceptable accuracy, avoiding excessive complexity.

This manuscript is a resubmission of an earlier submission. The following is a list of the peer review reports and author responses from that submission.

Round 1

Reviewer 1 Report

The system is clearly modeled and the final objective is well defined. However, the reason for adopting neural network is not so clear. The objective may be achieved by conventional analytical and/or statistical methods because the system is clearly modeled. As you have mentioned in the paper, the state of the art neural networks still have some drawbacks. You need sufficient amount of data for training and reverse engineering of the trained network is very limited. How are you going to obtain the training data in the real application? Please add some discussions on application of neural networks on your study.

Reviewer 2 Report

This is an interesting research problem, but it is not well articulated and presented in the manuscript.

An overview of the proposed system structure (Fig. 1, 2) has to be presented in the form of a flow chart and mathematical formulation. As is now it is not  clear for the reader.

The same applies to models presented via Simulink models (Fig. 3, 4, 5).

These Simulink models need to be presented subsequent to the flow-charts if they present any significant scientific value. As presented now, they don't present any considerable scientific value. 

The presented Simulink models shall accompany adequately formulated system equations. Otherwise, they don't contribute any significant value to the content of the paper. For instance, it is not clear system formulations under subsystems.

Core focus and novelty of the paper are the application of ANN, but it is not well articulated how ANN application was designed and applied in predicting health status and behavior offset of EMA. 

Since most of the obtained results (Target vs. Outputs) demonstrate linear relationship and thus, it not convincing whether the choice of ANN is the best or better one. As we all know ANN is computationally costly method. Some benchmarking of other methods in comparison to ANN method is needed. Since this study is only done numerically (theoretical calculations) and not backed with the experimental study results.

Reviewer 3 Report

1. There are many explanations that need to be reduced. 2. Please focus on the main discussion. 3. Simulation can be improved. 4. Use new articles in the introduction.